# Unveiling the Power of Multiple Gossip Steps: A Stability-Based Generalization Analysis in Decentralized Training

**Qinglun Li[1], Yingqi Liu[2], Miao Zhang[1], Xiaochun Cao[2], Quanjun Yin[1,*], Li Shen[2,3,*]**

[1]State Key Laboratory of Digital Intelligent Modeling and Simulation,
National University of Defense Technology, Changsha, 410073
[2]School of Cyber Science and Technology, Shenzhen Campus of Sun Yat-sen University, China
[3] Center for AI Theoretical Foundation and Systems, Shenzhen Loop Area Institute, China
liqinglun@nudt.edu.cn,yin_quanjun@163.com,mathshenli@gmail.com

## Abstract

Decentralized training removes the centralized server, making it a communication-efficient approach that can significantly improve training efficiency, but it often suffers from degraded performance compared to centralized training. Multi-Gossip Steps (MGS) serve as a simple yet effective bridge between decentralized and centralized training, significantly reducing experiment performance gaps. However, the theoretical reasons for its effectiveness and whether this gap can be fully eliminated by MGS remain open questions. In this paper, we derive upper bounds on the generalization error and excess error of MGS using stability analysis, systematically answering these two key questions. 1). *Optimization Error Reduction*: MGS reduces the optimization error bound at an exponential rate, thereby exponentially tightening the generalization error bound and enabling convergence to better solutions. 2). *Gap to Centralization*: Even as MGS approaches infinity, a non-negligible gap in generalization error remains compared to centralized mini-batch SGD ($\mathcal{O}(T^{\frac{c\beta}{c\beta+1}}/nm)$ in centralized and $\mathcal{O}(T^{\frac{2c\beta}{2c\beta+2}}/nm^{\frac{1}{2c\beta+2}})$ in decentralized). Furthermore, we provide the first unified analysis of how factors like learning rate, data heterogeneity, node count, per-node sample size, and communication topology impact the generalization of MGS under non-convex settings without the bounded gradients assumption, filling a critical theoretical gap in decentralized training. Finally, promising experiments on CIFAR datasets support our theoretical findings.

## 1 Introduction

Recently, decentralized training [1, 2] has emerged as a promising alternative to centralized training, which suffers from challenges like high communication overhead [3], single point of failure [4], and privacy risks [5]. In contrast, decentralized training eliminates the central server, offering stronger privacy protection [6], faster model training [7, 2], and robustness to slow client devices [8], making it an increasingly popular method [4, 7].

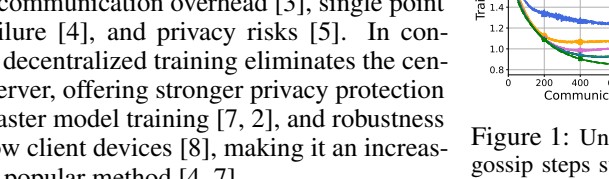
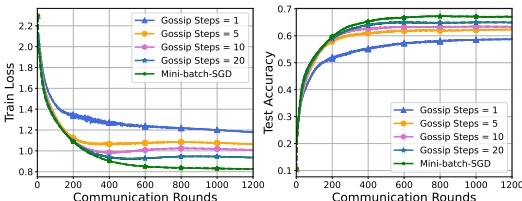

Figure 1: Under ring topology, DSGD-MGS with 20 gossip steps still shows significant performance gaps versus Mini-batch SGD in both training loss and test accuracy (LeNet on CIFAR-10, Dir 0.3, 50 nodes).

---

[*]Corresponding Authors

39th Conference on Neural Information Processing Systems (NeurIPS 2025).

However, despite the aforementioned advantages of decentralized training, some works [9, 10, 11] have pointed out that decentralized training methods underperform compared to centralized training methods in terms of model performance. Therefore, improving the performance of decentralized training models remains an important research question. Multiple Gossip Steps (MGS) [12, 13], as a simple yet effective method to enhance the performance of decentralized training models, has been experimentally proven to significantly improve the efficiency and performance of decentralized training [14, 15, 16, 17]. Even under communication compression, MGS continues to demonstrate its advantages in performance improvement [18].

Despite the substantial empirical benefits of MGS, the underlying theoretical understanding of its efficacy and its potential to eliminate the performance gap with centralized training remain critical open questions. Specifically, two key issues need to be addressed:

> (1) Why is MGS effective in improving model performance?
> (2) Can decentralized training ultimately match or even surpass the performance of centralized training by increasing the number of gossip steps?

To answer these open questions, we aim to theoretically explain how MGS works and how it affects model generalization. Using stability analysis, we find upper limits for the generalization error and excess error of MGS, giving systematic theoretical answers to these two main questions.

For Question 1, our theoretical analysis shows that MGS can reduce the optimization error bound at an exponential rate. This reduction in optimization error directly leads to an exponential reduction in the generalization error (as shown in Theorem 2, 3, and Remark 2), enabling the model to find better solutions. This relationship clearly explains why MGS effectively improves model performance. As illustrated in Figure 1, when the number of gossip steps is increased from 1 to 5, there is a significant reduction in the training loss (indicating reduced optimization error), and the test accuracy (measuring generalization) also shows a noticeable improvement. Furthermore, this improvement tends to diminish almost linearly as the number of gossip steps increases exponentially, consistent with the exponential decay in our theory findings.

For Question 2, our further analysis shows that even with a very large number of gossip steps, a basic difference in generalization error remains between decentralized DSGD-MGS and centralized mini-batch SGD.

Specifically, when the number of gossip steps becomes extremely large, the generalization error bound for DSGD-MGS becomes at most $\mathcal{O}(T^{\frac{2c\beta}{2c\beta+2}}/nm^{\frac{1}{2c\beta+2}})$. However, this is still noticeably larger than the centralized mini-batch SGD bound of $\mathcal{O}(T^{\frac{c\beta}{c\beta+1}}/nm)$, highlighting a lasting difference in how it scales with the number of clients $m$ (because $1/m < 1/m^{\frac{1}{2c\beta+2}}$ when $m > 1$). This theoretical observation reveals a basic constraint: decentralized training cannot fully achieve the generalization performance of centralized training solely by increasing the number of MGS steps. Experiments shown in Figure 1 support this conclusion, indicating that even with 20 gossip steps, DSGD-MGS still performs worse than centralized mini-batch SGD in the same settings.

Moreover, we are the first to provide a theoretical framework to understand how critical factors, including learning rate, data heterogeneity, number of nodes, sample size per node, and communication topology, jointly influence the generalization performance of MGS (see Reamrk 2-9). Remarkably, we also eliminate the bounded gradient assumption in the non-convex condition. This work enhances our understanding of the challenges in decentralized learning and provides theoretical insights for hyperparameters to better model generalization. Finally, extensive experiments on CIFAR datasets further validate our theoretical results. The main contributions of this paper can be summarized as follows:

- Theoretically elucidating the mechanism by which MGS enhances the generalization performance of decentralized training models through an exponential reduction in optimization error.

- Revealing that even with sufficient gossip communication, a theoretical gap in generalization error remains between MGS and centralized training, and this gap cannot be eliminated by MGS alone.

- Establishing, for the first time under non-convex and without the bound gradient assumption, a unified framework analyzing factors impacting the MGS generalization performance (i.e., learning rate, data heterogeneity, number of nodes, sample size, and topology), thereby addressing a significant gap in existing theoretical frameworks.

- Validating our theoretical findings through empirical experiments on the CIFAR datasets.

These findings provide new theoretical insights and practical implications for understanding and improving decentralized learning algorithms.

## 2 Related Works

This section reviews the current theoretical understanding and challenges in decentralized training, along with the evolution and impact of MGS. Moreover, at the end of each subsection, we highlight the existing gaps and open questions within these areas to position the contributions of this paper.

**Theoretical Analysis of D-SGD.** Decentralized learning has attracted significant research interest due to its potential for enhanced privacy, communication efficiency, and scalability [7, 5, 6, 8]. Early theoretical studies primarily focused on the convergence analysis of D-SGD, examining the number of iterations or communication rounds needed to reach an $\epsilon$-accurate solution [7, 18, 19]. More recently, attention has shifted towards understanding the generalization performance of these algorithms. Sun et al. [20] were the first to analyze the generalization performance of D-SGD using uniform stability, later extending their results to asynchronous D-SGD [21]. However, these analyses assumed *homogeneous data and bounded gradients*. Zhu et al. [22] further studied the impact of communication topology on the generalization error of D-SGD, with their generalization bounds later improved by [11], but they also relied on the same assumptions. More recently, Ye et al. [23] analyzed the generalization behavior of D-SGD under heterogeneous data, but their analysis was limited to *strongly convex* loss functions. Overall, current D-SGD theories still lack a unified framework that comprehensively accounts for all key algorithm parameters (e.g., data heterogeneity, non-convex loss function, topology, etc.).

**MGS in Decentralized Training.** Multiple Gossip Steps (MGS) [24, 12] is a technique that improves consensus by allowing multiple rounds of local communication. When integrated into decentralized algorithms, MGS not only enhances generalization performance but also accelerates convergence [25]. Additionally, Yuan et al. [19] showed that MGS can reduce the adverse effects of data heterogeneity, a finding supported by other studies [26, 16]. Li et al. [27] found that MGS can significantly improve algorithm accuracy. In the field of decentralized federated learning, Shi et al. [16] incorporated MGS into their DFedSAM algorithm, significantly improving its generalization performance experimentally. Notably, MGS alone can achieve optimal convergence rates in non-convex settings [19] without relying on more complex techniques like gradient tracking [28], quasi-global momentum [29], or adaptive momentum [30]. However, these studies have largely overlooked the question of why MGS is effective from a generalization perspective, with these advantages demonstrated mainly through empirical results, leaving a significant gap in the theoretical understanding of MGS.

## 3 Background

In this section, we first present some fundamental definitions required for stability analysis, including population risk, empirical risk, generalization error, excess error, and $l_2$ on-average model stability. Subsequently, we introduce a key lemma that establishes the relationship between the generalization error bound and $l_2$ on-average model stability.

### 3.1 Stability and Generalization in Decentralized Learning

We consider the general statistical learning setting, adapted to a decentralized framework with $m$ agents[2]. Each agent $k$ observes data points drawn from a local distribution $\mathcal{D}_k$ with support $\mathcal{Z}$. The goal is to find a global model $\theta \in \mathbb{R}^d$ that minimizes the *population risk*, defined as:

$$R(\theta) \triangleq \frac{1}{m} \sum_{k=1}^{m} l_k(\theta) \triangleq \frac{1}{m} \sum_{k=1}^{m} \mathbb{E}_{Z \sim \mathcal{D}_k} [\ell(\theta; Z)] \,,$$

where $\ell$ is some loss function. We denote by $\theta^\star$ a global minimizer of the population risk, i.e., $\theta^\star \in \arg\min_\theta R(\theta)$.

---

[2]In this paper, the terms node, agent, and client are used interchangeably.

Although the population risk $R(\theta)$ is not directly computable, we can instead evaluate an empirical counterpart using $m$ local datasets $S \triangleq (S_1, \ldots, S_m)$, where $S_k = \{Z_{1k}, \ldots, Z_{nk}\}$ represents the dataset of agent $k$, with each sample $Z_{ik}$ drawn from the local distribution $\mathcal{D}_k$. For simplicity, we assume that each local dataset has the same size $n$, though our analysis can be extended to the heterogeneous case. The resulting *empirical risk* is given by:

$$R_S(\theta) \triangleq \frac{1}{m} \sum_{k=1}^{m} R_{S_k}(\theta) \triangleq \frac{1}{mn} \sum_{k=1}^{m} \sum_{i=1}^{n} \ell(\theta; Z_{ik}) \ .$$

One of the most well-known and extensively studied estimators is the empirical risk minimizer, defined as $\widehat{\theta}_{\mathrm{ERM}} \triangleq \arg\min_\theta R_S(\theta)$. However, in most practical scenarios, directly computing this estimator is infeasible. Instead, one typically employs a potentially random *decentralized optimization* algorithm $A$, which takes the full dataset $S$ as input and returns an approximate minimizer $A(S) \in \mathbb{R}^d$ for the empirical risk $R_S(\theta)$.

In this setting, the expected *excess risk* $R(A(S)) - R(\theta^\star)$ can be upper-bounded by the sum of the (expected) *generalization error* ($\epsilon_{\mathrm{gen}}$) and the (expected) *optimization error* ($\epsilon_{\mathrm{opt}}$) [23, 11]:

$$\mathbb{E}_{A,S}[R(A(S)) - R(\theta^\star)] \leq \epsilon_{\mathrm{gen}} + \epsilon_{\mathrm{opt}} \tag{3.1}$$

where $\epsilon_{\mathrm{gen}} \triangleq \mathbb{E}_{A,S}[R(A(S)) - R_S(A(S))]$ and $\epsilon_{\mathrm{opt}} \triangleq \mathbb{E}_{A,S}[R_S(A(S)) - R_S(\widehat{\theta}_{\mathrm{ERM}})]$. This work focuses on controlling the expected generalization error $\epsilon_{\mathrm{gen}}$, for which a common approach is to use the stability analysis of the algorithm $A$.

Contrary to a large body of works using the well-known *uniform stability* [31, 32], our analysis relies on the notion of *on-average model stability* [33], which has the advantage of removing the bounded gradient assumption [3, 34, 10] in our analysis, making the theoretical results more general. Below, we recall this notion, with a slight adaptation to the decentralized setting.

**Definition 1** ($l_2$ **on-average model stability**). *Let $S = (S_1, \ldots, S_m)$ with $S_k = \{Z_{1k}, \ldots, Z_{nk}\}$ and $\tilde{S} = (\tilde{S}_1, \ldots, \tilde{S}_m)$ with $\tilde{S}_k = \{\tilde{Z}_{1k}, \ldots, \tilde{Z}_{nk}\}$ be two independent copies such that $Z_{ik} \sim \mathcal{D}_k$ and $\tilde{Z}_{ik} \sim \mathcal{D}_k$. For any $i \in \{1, \ldots, n\}$ and $j \in \{1, \ldots, m\}$, let us denote by $S^{(ij)} = (S_1, \ldots, S_{j-1}, S_j^{(i)}, S_{j+1}, \ldots, S_m)$, with $S_j^{(i)} = \{Z_{1j}, \ldots, Z_{i-1j}, \tilde{Z}_{ij}, Z_{i+1j}, \ldots, Z_{nj}\}$, the dataset formed from $S$ by replacing the $i$-th element of the $j$-th agent's dataset by $\tilde{Z}_{ij}$. A randomized algorithm $A$ is said to be $l_2$ on-average model $\varepsilon$-stable if*

$$\mathbb{E}_{S,\tilde{S},A}\Big[\frac{1}{mn}\sum_{i=1}^{n}\sum_{j=1}^{m}||A(S) - A(S^{(ij)})||_2^2\Big] \leq \varepsilon^2 \ . \tag{3.2}$$

A key aspect of on-average model stability is that it can directly be linked to the generalization error, as shown in the following lemma.

**Lemma 1** (**Generalization via on-average model stability [33]**). *Let $A$ be $l_2$ on-average model $\varepsilon$-stable. Let $\gamma > 0$. Then, if $\ell(\cdot; z)$ is nonnegative and is $\beta$-smoothness for all $z \in \mathcal{Z}$, we have*

$$\epsilon_{gen} \leq \frac{1}{2mn\gamma} \sum_{i=1}^{n}\sum_{j=1}^{m} \mathbb{E}_{A,S}[||\nabla\ell(A(S); Z_{ij})||^2] + \frac{\beta + \gamma}{2mn} \sum_{i=1}^{n}\sum_{j=1}^{m} \mathbb{E}_{A,\tilde{A},S}[||A(S) - A(S^{(ij)})||^2]$$

In fact, we modified the proof of the lemma from Lei et al.[33], replacing the $R_S(A(S))$ on the right-hand side with a gradient $\mathbb{E}_{A,S}[||\nabla\ell(A(S); Z_{ij})||^2]$. This adjustment better captures the impact of data heterogeneity on the generalization error. With this lemma, obtaining the desired generalization bound reduces to controlling the $l_2$ on-average model stability of the decentralized algorithm $A$.

### 3.2 Decentralized SGD with Multiple Gossip Steps

In this paper, we focus on the widely-used Decentralized Stochastic Gradient Descent (D-SGD) algorithm [35, 7], which aims to find minimizers (or saddle points) of the empirical risk $R_S(\theta)$ in a fully decentralized manner. This algorithm relies on peer-to-peer communication between agents, with a graph representing which pairs of agents (or nodes) are able to interact. Specifically, the *communication topology* is captured by a gossip matrix $W \in [0, 1]^{m \times m}$ (see Definition 2), where $W_{jk} > 0$ indicates the weight that agent $j$ assigns to messages from agent $k$, and $W_{jk} = 0$ (no edge) implies that agent $j$ does not receive messages from agent $k$.

The D-SGD with Multiple Gossip Steps (DSGD-MGS) algorithm performs multiple gossip updates during the communication phase of the D-SGD algorithm, while all other computational components remain identical to D-SGD, as detailed in Algorithm 1. Specifically, the main procedure at time $t$ is divided into two steps:

- **Local Update Steps:** Each node independently and uniformly draws a training sample $Z_{I_k^t k}$ from its local dataset $S_k$. Based on the current model parameter $\theta_k^{(t)}$, it computes the gradient $\nabla \ell(\theta_k^{(t)}; Z_{I_k^t k})$ and performs gradient descent to obtain the initial point for Multiple Gossip Steps: $\theta_k^{(t,0)} = \theta_k^{(t)} - \eta_t \nabla \ell(\theta_k^{(t)}; Z_{I_k^t k})$, where $\eta_t$ denotes the step size.

---

**Algorithm 1** Decentralized SGD with MGS

1: **Input:** Initialize $\forall k$, $\theta_k^{(0)} = \theta^{(0)} \in \mathbb{R}^d$, iterations $T$, stepsizes $\{\eta_t\}_{t=0}^{T-1}$, weight matrix $W$, Multiple Gossip Steps $Q$.
2: **for** $t = 0, \ldots, T-1$ **do**
3:    **for** each node $k = 1, \ldots, m$ in parallel **do**
4:       Local Update Steps:
5:       Sample $I_k^t \sim \mathcal{U}\{1, \ldots, n\}$
6:       $\theta_k^{(t,0)} = \theta_k^{(t)} - \eta_t \nabla \ell(\theta_k^{(t)}; Z_{I_k^t k})$
7:       Multiple Gossip Steps:
8:       **for** $q = 0$ to $Q-1$ **do**
9:          $\theta_k^{(t,q+1)} = \sum_{l=1}^m W_{kl} \theta_l^{(t,q)}$
10:      **end for**
11:      $\theta_k^{(t+1)} = \theta_k^{(t,Q)}$
12:    **end for**
13: **end for**

---

- **Multiple Gossip Steps:** Each node exchanges information with its neighbors through $Q$ gossip averaging steps: $\theta_k^{(t,q+1)} = \sum_{l=1}^m W_{kl} \theta_l^{(t,q)}$. The resulting model parameter $\theta_k^{(t,q+1)}$ is then used as the initial point $\theta_k^{(t+1)}$ for the next Local Update Steps.

## 4 Generalization Analysis

In this section, we first introduce the Definition and Assumptions required for analyzing the generalization of the DSGD-MGS algorithm. We then present the upper bounds for the generalization error and excess error, followed by a detailed analysis of these bounds. Proofs for all Lemmas and Theorems can be found in the **Appendix** B.

### 4.1 Definition and Assumption

**Definition 2** (Gossip Matrix). *Let $W \in [0,1]^{n \times n}$ be a symmetric doubly stochastic matrix. This means that $W = W^\top$, and both the row sums and column sums of $W$ equal one, i.e., $W\mathbf{1} = \mathbf{1}$ and $\mathbf{1}^\top W = \mathbf{1}^\top$, where $\mathbf{1}$ is the vector of all ones. The eigenvalues of $W$ are ordered as $1 = |\lambda_1(W)| > |\lambda_2(W)| \geq \cdots \geq |\lambda_n(W)|$. The spectral gap of $W$, denoted by $\delta$, is defined as $\delta := 1 - |\lambda_2(W)| \in (0,1)$.*

**Assumption 1.** ($\beta$-smoothness). *The loss function $\ell$ is $\beta$-smooth i.e. $\exists \beta > 0$ such that $\forall \theta, \theta' \in \mathbb{R}^d, z \in \mathcal{Z}, \|\nabla \ell(\theta; z) - \nabla \ell(\theta'; z)\|_2 \leq \beta \|\theta - \theta'\|_2$.*

**Assumption 2.** (Bounded Stochastic Gradient Noise). *There exists $\sigma^2 > 0$ such that $\mathbb{E}_{Z_{i,j}} \|\nabla \ell(\theta; Z_{i,j}) - \nabla R_{\mathcal{S}_j}(\theta)\|^2 \leq \sigma^2$, for any agent $j \in [m]$ and $\theta \in \mathbb{R}^d$.*

**Assumption 3.** (Bounded Heterogeneity). *There exists $\xi^2 > 0$ such that $\frac{1}{m} \sum_{k=1}^m \|\nabla R_{S_k}(\theta) - \nabla R_S(\theta)\|^2 \leq \xi^2$, for any $\theta \in \mathbb{R}^d$.*

Using the property $\beta$-smoothness of $\ell(\theta; z)$, it is straightforward to show that $\ell_k(\theta) = \mathbb{E}_{Z \sim \mathcal{D}_k}[\ell(\theta; Z)]$ and $R_{S_k}(\theta) = \frac{1}{n} \sum_{i=1}^n \ell(\theta; Z_{ik})$ also satisfy the property $\beta$-smoothness.

**Remark 1.** *Definition 2 stipulates that the communication topology must be a doubly stochastic matrix, which appears in many decentralized optimization works [7, 34, 11, 18, 3]. Assumption 1 specifies that the loss function is smooth, which is often used in optimization and generalization studies under non-convex settings [36, 10, 37, 38, 39, 40]. Assumption 2 states that the stochastic gradients of the samples are bounded, and Assumption 3 bounds the heterogeneity of the data. These assumptions are frequently used in the convergence analysis of many works [36, 3, 16, 37], and we will employ them in this paper to analyze the stability and generalization of DSGD-MGS.*

### 4.2 Generalization Error and Excess Error of DSGD-MGS

Due to its fully decentralized structure, DSGD-MGS produces $m$ distinct outputs, $A_1(S) \triangleq \theta_1^{(T)}, \ldots, A_m(S) \triangleq \theta_m^{(T)}$, one for each agent. As a result, the stability and generalization anal-

ysis that follows will focus on these individual outputs, rather than a single global output $A(S)$ as described in Section 3.1. Denote by $A_k(S) = \theta_k^{(T)}$ and $A_k(S^{(ij)}) = \theta_k^{(T)}(i,j)$, the final iterates of agent $k$ for DSGD-MGS run over two data sets $S$ and $S^{(ij)}$ that differ only in the $i$-th sample of agent $j$. To obtain a tighter upper bound for the non-convex case, we modify Lemma 1 by introducing a variable $t_0$, resulting in the following key lemma, which transforms the computation of the generalization error upper bound $\epsilon_{\text{gen}}$ into the computation of the stability upper bound.

**Lemma 2.** *Assume the loss function $\ell(\cdot, z)$ is nonnegative and bounded in $[0, 1]$, and that Assumptions 1 hold. For all $i = 1, \ldots, n$ and $j = 1, \ldots, m$, let $\{\theta_k^{(t)}\}_{t=0}^T$ and $\{\tilde{\theta}_k^{(t)}(i,j)\}_{t=0}^T$, the iterates of agent $k = 1, \ldots, m$ for DSGD-MGS run on $S$ and $S^{(ij)}$ respectively. Then, for every $t_0 \in \{0, 1, \ldots, T\}$ we have:*

$$|\mathbb{E}_{A,S}[R(A_k(S)) - R_S(A_k(S))]|$$
$$\leq \frac{t_0}{n} + \underbrace{\frac{\gamma + \beta}{2mn} \sum_{i=1}^n \sum_{j=1}^m \mathbb{E}[\delta_k^{(T)}(i,j)|\delta^{(t_0)}(i,j) = \mathbf{0}]}_{I_1 : l_2 \text{ on-average model stability}} + \underbrace{\frac{1}{2mn\gamma} \sum_{i=1}^n \sum_{j=1}^m \mathbb{E}[\|\nabla \ell(A_k(S); Z_{ij})\|^2]}_{I_2 : \text{Related to optimization error}}$$

*where $\delta^{(t)}(i,j)$ is the vector containing $\forall k = 1, \ldots, m$, $\delta_k^{(t)}(i,j) = \|\theta_k^{(t)} - \tilde{\theta}_k^{(t)}(i,j)\|_2^2$.*

According to Lemma 2, to compute the generalization error $\epsilon_{\text{gen}}$, We need to calculate the $l_2$ on-average model stability ($I_1$) and the gradient related to the optimization error ($I_2$). Below, we first provide the stability upper bound, followed by the optimization error upper bound.

**Upper bound of $I_1$:** For a fixed couple $(i,j)$, we are first going to control the vector $\Delta^{(t)} = \frac{1}{mn} \sum_{i,j} \Delta^{(t)}(i,j)$, where $\Delta^{(t)}(i,j) \triangleq \mathbb{E}[\delta^{(t)}(i,j)|\delta^{(t_0)}(i,j) = \mathbf{0}]$. When it is clear from context, we simply write $\tilde{\theta}_k^{(t)}(i,j) = \tilde{\theta}_k^{(t)}$. Next, we provide the upper bound of the $l_2$ on-average model stability for the DSGD-MGS algorithm.

**Theorem 1** (**Stability for the DSGD-MGS**). *As in the conditions of Lemma 2, then the following holds:*

$$\frac{1}{mn} \sum_{i=1}^n \sum_{j=1}^m \mathbb{E}[\delta_k^{(T)}(i,j)|\delta^{(t_0)}(i,j) = \mathbf{0}] \leq \frac{8e\sqrt{2\beta}c^2}{(1 + 2c\beta)nmt_0} \left(\frac{T}{t_0}\right)^{2c\beta}$$

**Upper bound of $I_2$:** Let $\bar{G} = \frac{1}{mn} \sum_{i,j} \mathbb{E}[\|\nabla \ell(\theta_k^{(T)}; Z_{ij})\|^2]$. According to the Assumptions 2 and 3, the following inequality holds:

$$\bar{G} = \frac{1}{mn} \sum_{i,j} \mathbb{E}[\|\nabla \ell(\theta_k^{(T)}; Z_{ij})\|^2] = \frac{1}{mn} \sum_{i,j} \mathbb{E}[\|\nabla \ell(\theta_k^{(T)}; Z_{ij}) \pm \nabla R_{S_k}(\theta_k^{(T)}) \pm \nabla R_S(\theta_k^{(T)})\|^2]$$

$$\leq 3\sigma^2 + 3\xi^2 + 3\mathbb{E}[\|\nabla R_S(\theta_k^{(T)})\|^2]$$

Since $\ell$ satisfies the $\beta$-smoothness property, it is straightforward to show that $R_S(\theta_k^{(T)})$ also satisfies the $\beta$-smoothness property. Consequently, $R_S(\theta)$ also satisfies the self-bounding property in Lemma 3 (see the **Appendix** B), i.e., $\|\nabla R_S(\theta)\|^2 \leq 2\beta R_S(\theta)$. Then, we have

$$\bar{G} \leq 3\sigma^2 + 3\xi^2 + 6\beta \mathbb{E}_S[R_S(\theta_k^{(T)})] \tag{4.1}$$

Next, we will focus on bounding $\mathbb{E}_S[R_S(\theta_k^{(T)})]$. According to the results from [18, Theorem 1] (see Lemma 2 in the **Appendix** B), we have the following theorem:

**Theorem 2** (**Optimization error of DSGD-MGS**). *Let $\Delta^2 := \max_{\theta^* \in \mathcal{X}^*} \sum_{k=1}^m \|\nabla R_{S_k}(\theta^*)\|^2$, $R_0 := R_S(\theta^{(0)}) - R_S^*$, where $\mathcal{X}^* = \arg\min_\theta R_S(\theta)$ and $R_S^* = R_S(\widehat{\theta}_{ERM})$. Suppose Assumptions 1 and Polyak-Łojasiewicz (PL) condition (see Assumption 4 in the **Appendix**) hold. Define*

$$Q_0 := \log(\bar{\rho}/46)/\log\left(1 - \frac{\delta\tilde{\gamma}}{2}\right), \bar{\rho} := 1 - \frac{\mu}{m\beta}, \tilde{\gamma} = \frac{\delta}{\delta^2 + 8\delta + (4 + 2\delta)\lambda_{\max}^2(I - W)}.$$

Then, if the nodes are initialized such that $\theta_k^Q = 0$, for any $Q > Q_0$ after $T$ iterations the iterates of DSGD-MGS with $\eta_t = \frac{1}{\beta}$ satisfy

$$\mathbb{E}_S[R_S(\theta_k^{(T)})] - R_S^* = \mathcal{O}\left(\frac{\Delta^2 e^{-\frac{\delta\tilde{\gamma}Q}{4}}}{1 - \bar{\rho}} + \left[1 + \frac{\beta}{\mu\bar{\rho}}\left(1 + e^{-\frac{\delta\tilde{\gamma}Q}{4}}\right)\right]R_0\rho^T\right). \quad (4.2)$$

Here, $\delta$ represents the spectral gap of $W$, and $\rho \triangleq 1 - \delta = |\lambda_2(W)|$ is defined in definition 2.

By combining Equation (4.1) with Theorem 2, we obtain the upper bound for $\bar{G}$.

$$\bar{G} = \mathcal{O}(\sigma^2 + \delta^2 + R_S^*) + \mathcal{O}\left(\frac{\beta\Delta^2 e^{-\frac{\delta\tilde{\gamma}Q}{4}}}{1 - \bar{\rho}} + \left[1 + \frac{\beta}{\mu\bar{\rho}}\left(1 + e^{-\frac{\delta\tilde{\gamma}Q}{4}}\right)\right]R_0\beta\rho^T\right) \quad (4.3)$$

**Generalization Bound for DSGD-MGS:** With the above Theorem 1 & 2, we can derive the generalization error upper bound for DSGD-MGS.

**Theorem 3** (**Generalization error of DSGD-MGS**). *Based on Lemma 2, Theorem 1 and Theorem 2, and assuming that Assumptions 1-3 hold, let the learning rate satisfy $\eta_t \leq \frac{c}{t+1}$ for some constant $c > 0$. We derive the following result by appropriately selecting $t_0$ and $\gamma$:*

$$|\mathbb{E}_{A,S}[R(A_k(S)) - R_S(A_k(S))]|$$

$$\leq \frac{2c\beta + 3}{(n(2c\beta + 1))^{\frac{2c\beta+2}{2c\beta+3}}}\left(\frac{2\bar{G}e\sqrt{2\beta}c^2 T^{2c\beta}}{m}\right)^{\frac{1}{2c\beta+3}} + \frac{2c\beta + 2}{n(2c\beta + 1)}\left(\frac{4\beta e\sqrt{2\beta}c^2 T^{2c\beta}}{m}\right)^{\frac{1}{2c\beta+2}}$$

*where the expression for $\bar{G}$ is given in Equation (4.3).*

**Remark 2** (**Optimization Error Reduction**). *As shown in Theorem 3, the generalization error bound obtained via $l_2$ on-average model stability is closely related to the optimization error $\bar{G}$. Analyzing the MGS-related terms reveals that increasing the number of MGS steps $Q$ reduces $\bar{G}$, thereby tightening the generalization error bound. Moreover, a more detailed analysis shows that the reduction in the generalization error bound is exponential, specifically on the order of $\mathcal{O}(e^{-\frac{\delta\gamma Q}{4}})$, indicating that even a small increase in $Q$ can lead to significant gains. This observation will also be validated in the experimental section 5.2.*

**Remark 3** (**Gap to Centralization**). *As indicated by Theorem 3, by letting $Q$ approach infinity, we can derive the limiting generalization error bound, which helps address whether DSGD-MGS with sufficiently many steps can effectively approximate centralized mini-batch SGD. The answer is no, because the resulting bound is at most $\mathcal{O}\left(T^{\frac{2c\beta}{2c\beta+2}}/nm^{\frac{1}{2c\beta+2}}\right)$, which still differs in terms of node count $m$ and per-node data size $n$ from the bound $\mathcal{O}\left(T^{\frac{c\beta}{c\beta+1}}/mn\right)$ established for centralized mini-batch SGD based on uniform stability in [11, 41]. Therefore, this gap persists unless the number of nodes or the data size per node is significantly increased. As illustrated in Figure 1.*

**Remark 4** (**Related to the Optimization Error**). *Compared to prior works on the generalization error of D-SGD [42, 41, 11, 22], which rely on Lipschitz assumptions for the loss function, our approach removes this assumption, allowing for a more explicit connection between optimization error and generalization error. In those works, the Lipschitz assumption effectively absorbs optimization-related quantities (e.g., gradients) into a Lipschtiz constant, obscuring this relationship. In contrast, our work removes the Lipschitz assumption, making the relationship between generalization and optimization errors more explicit. Our results show that reducing optimization error can also decrease generalization error to some extent, which explains the common observation that as training progresses, both the training error decreases and the model's performance on the validation set improves.*

**Remark 5** (**Influential Factors of the Generalization Error for DSGD-MGS**). *When the model, loss function, and dataset are fixed, parameters like the smoothness $\beta$, gradient noise $\sigma$, and data heterogeneity $\delta$ are also fixed. In this case, to reduce the generalization error bound according to the upper bound in Theorem 3, the following strategies are effective: 1) Increase the data size per node $n$; 2) Increase the number of nodes $m$; 3) Increase the MGS step count $Q$; 4) Reduce the distance between the optimal point and the initial point $R_0$; 5) Use a communication topology with a larger spectral gap $\delta$ (which implies a smaller $\rho$); 6) Decrease the learning rate $c$. The first five are straightforward, while the sixth is recommended because the number of iterations $T$ is usually large, making*

$T^{\frac{2c\beta}{2c\beta+2}}$ *the dominant term in the bound. Reducing* $c$ *can significantly reduce this term. Additionally, if the choice of dataset is flexible, selecting one that is as close to i.i.d. as possible is beneficial, as a larger data heterogeneity parameter* $\xi$ *will generally increase the generalization error bound.*

**Remark 6** (**Innovation in Generalization Error Bounds**)**.** *Our work introduces* $l_2$ *on-average model stability to deriving generalization error bounds for decentralized algorithms, characterized by the following key innovations: 1) Removal of Lipschitz Assumption: Unlike previous proofs based on uniform stability [41, 20, 11, 10, 9, 43], our approach removes the Lipschitz assumption on the loss function (which implicitly bounds the gradient), allowing the relationship between optimization error and generalization error to become more explicit. 2) Explicit Role of Optimization Error: We establish, for the first time, a direct connection between the optimization error and generalization error of the D-SGD algorithm, revealing that reducing the optimization error also decreases the generalization error, which aligns better with observed training dynamics. 3) Exponential MGS Benefit: Our bounds demonstrate that the impact of MGS on reducing generalization error is exponential, highlighting the significant gains achievable with a moderate number of MGS steps. 4) Quantification of Heterogeneity Impact: Ye et al.[23] were the first to theoretically reveal that data heterogeneity can degrade the generalization bound of the D-SGD algorithm under the strongly convex setting. Building on this, we take a further step by providing a precise characterization of how data heterogeneity affects generalization in the non-convex setting, filling a critical gap in existing theoretical analyses.*

**Theorem 4** (**Excess Error of DSGD-MGS**)**.** *Under the same conditions and notation as Theorems 3 and 2, and based on the decomposition of excess error in Equation (3.1), the optimization error bound (Equation 4.2), and the generalization error bound (Theorem 3), we obtain the following upper bound for the excess error.*

$$\mathbb{E}_{A,S}[R(A(S)) - R(\theta^\star)] = \mathcal{O}\Bigg(\frac{\Delta^2 e^{-\frac{\delta\gamma Q}{4}}}{1-\bar{\rho}} + \left[1 + \frac{\beta}{\mu\bar{\rho}}\left(1 + e^{-\frac{\delta\gamma Q}{4}}\right)\right]R_0\rho^T \tag{4.4}$$
$$+ \frac{1}{n^{\frac{2c\beta+2}{2c\beta+3}}}\left(\frac{\bar{G}\beta^{\frac{3}{2}}c^3 T^{2c\beta}}{m}\right)^{\frac{1}{2c\beta+3}} + \frac{1}{n}\left(\frac{\beta^{\frac{3}{2}}c^2 T^{2c\beta}}{m}\right)^{\frac{1}{2c\beta+2}}\Bigg)$$

**Remark 7** (**The difference of conclusions obtained from excess error and generalization error**)**.** *Since the excess error can be decomposed as* $\mathbb{E}_{A,S}[R(A(S)) - R(\theta^\star)] \le \epsilon_{gen} + \epsilon_{opt}$, *most conclusions about the generalization error also apply to the excess error (see Remark 5). The only key difference lies in the choice of learning rate. For* $\epsilon_{gen}$, *a smaller learning rate (i.e., smaller* $c$*) is preferred, as* $\epsilon_{gen}$ *is dominated by the term* $\mathcal{O}(T^{\frac{2c\beta}{2c\beta+2}})$, *meaning that reducing* $c$ *significantly reduces this term and hence the generalization error. However, this is not the case for* $\epsilon_{opt}$. *Prior work on the convergence of D-SGD [7] shows that* $\epsilon_{opt} = \mathcal{O}\left(\frac{R_0}{T\eta}\right)$, *indicating that an excessively large learning rate increases* $\epsilon_{opt}$, *thereby undermining convergence. Thus, the choice of learning rate involves a trade-off between minimizing generalization error and maintaining convergence, a conclusion that will be confirmed in the Experimental Section A.2.*

**Remark 8.** (***On the Technical Role of the PL Condition***)**.** *Our analysis of the generalization error requires bounding the expected squared gradient norm at the final iterate, denoted as* $\bar{G}$. *However, establishing a tight upper bound for the final iterate's gradient in non-convex decentralized optimization remains a challenging frontier problem. While recent advances have been made in last-iterate convergence analysis (e.g., [44]), existing results either do not incorporate the MGS mechanism or provide bounds only on the function value gap, which are insufficient for directly bounding* $\bar{G}$. *To bridge this gap, we adopt the Polyak-Łojasiewicz (PL) condition. This is a standard approach in the literature (e.g., [34]) used to connect the squared gradient norm with the function value gap. This technical choice is deliberate and crucial, as a tight upper bound on the function value gap under the MGS setting is available [18]. Consequently, the PL condition enables us to derive some of the first fine-grained, MGS-aware generalization bounds that explicitly link the generalization error to key algorithmic hyperparameters, including the number of MGS steps (Q), communication topology, and learning rate. This provides concrete, quantitative insights that significantly advance beyond high-level bounds, such as the classic* $\mathcal{O}(1/T)$ *analysis provided by L2-stability [33]. Therefore, the reliance on the PL condition reflects the current theoretical limits in non-convex last-iterate analysis rather than a fundamental limitation of our stability framework. Our framework is modular: should future research provide a direct, assumption-free upper bound for* $\bar{G}$ *in the MGS setting, our generalization bounds can be immediately strengthened by replacing this component. A more detailed discussion is provided in the* ***Appendix D.4.***

**Remark 9.** *All the above discussions are also solid to $\bar{\theta}^{(T)} \triangleq \frac{1}{m}\sum_{k=1}^{m}\theta_k^{(T)}$. In addition, our theoretical results apply to decentralized topologies other than the fully connected case. When the topology becomes fully connected, the iterative update reduces to the centralized setting. For detailed analysis, please refer to the **Appendix**. For detailed proof, please refer to **Appendix D**. Additionally, we provide a **consensus error analysis** to further illustrate the behavior of MGS in both finite and infinite regimes (detailed discussion provided in **Appendix C**). Furthermore, we extend our theoretical analysis to the case involving batch size b. The detailed proofs and analyses are provided in the **Appendix D.2**.*

## 5 Experiment

In this section, we present extensive experiments to validate our theoretical findings. We first describe the experimental setup, followed by the empirical results and corresponding analysis. Due to space constraints, the experimental validation of excess error is presented in **Appendix** A.2. Furthermore, we conduct an in-depth exploration of the subtle relationship between mini-batch size and (Q) on the CIFAR-100 dataset, providing practitioners with insights for achieving higher performance. Detailed analyses and discussions can be found in the **Appendix** D.3.

### 5.1 Empirical Setup

We conduct experiments on the CIFAR-10 dataset [45] with a Dirichlet distribution (non-IID, $\alpha = 0.3$) using LeNet to validate the excess error and generalization error of DSGD-MGS. To examine the impact of key hyperparameters, we follow the study by Hardt et al.[41] and investigate the weight distance ($\sum_{i=1}^{n}\sum_{j=1}^{m}||\theta_j^{(t)} - \tilde{\theta}_j^{(t)}||_2^2$) and the loss distance ($R(\bar{\theta}^{(t)}) - R_S(\bar{\theta}^{(t)})$) when replacing only one data point in the training dataset. We primarily validate the experimental performance of key parameters in the DSGD-MGS algorithm, such as communication topology, the number of MGS steps, and the total number of clients. For fairness, when exploring one parameter, all other parameters are kept at the same settings. Further implementation details are provided in **Appendix** A.1.

### 5.2 Experimental Validation of Generalization Error.

As shown in Figure 2, subplots (a) and (b) respectively illustrate the weight distance and loss distance for different parameter settings of the DSGD-MGS algorithm on the perturbed dataset. Overall, both weight distance and loss distance exhibit the same power-law behavior as our theoretical bound $\mathcal{O}(T^{\frac{2c\beta}{2c\beta+2}})$ (see Theorem 3). Additionally, within each column of Figure 2 (corresponding to the same parameter setting), these two metrics follow similar trends, confirming the validity of Lemma 1 [33], which states that the generalization error can indeed be captured by the stability bound.

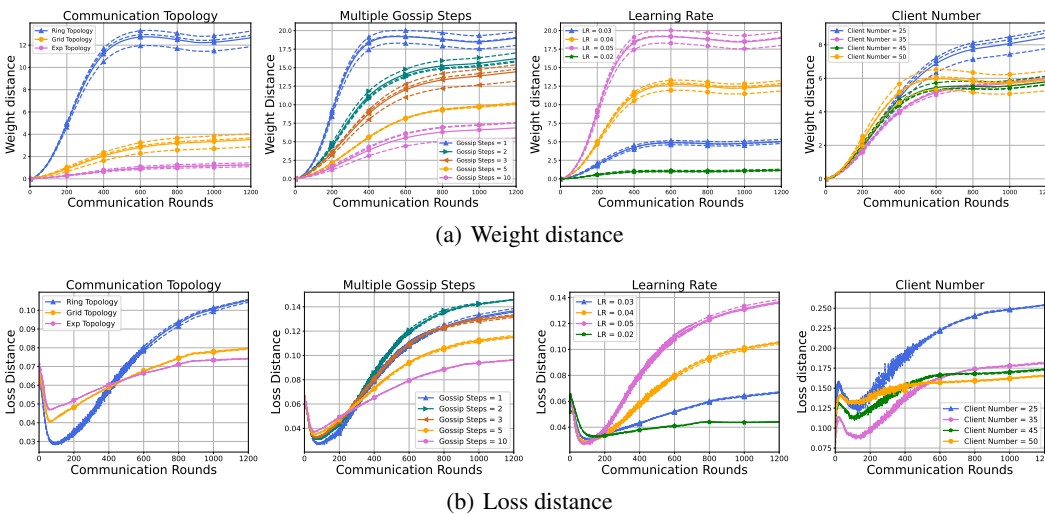

(a) Weight distance

(b) Loss distance

Figure 2: A comparison of the $l_2$ weight distance and Loss distance (i.e. test loss - train loss) for the DSGD-MGS algorithm on the cifar10 dataset.

From subplots (a) and (b) in Figure 2, we can observe the following patterns: 1) ***Using a communication topology with a smaller spectral gap*** (i.e., a larger $\rho$ in Theorem 3) leads to lower generalization error. 2) ***Increasing the number of MGS effectively reduces the generalization error.*** For example, in terms of weight distance (Figure 2 (a)), setting $MGS = 5$ reduces the weight distance to roughly half of that with $MGS = 1$. 3) ***Smaller learning rates help reduce generalization error***, consistent with the findings in [10] on decentralized federated learning. 4) ***A larger client number (i.e., $m$ in Theorem 3) also helps reduce generalization error***, reflecting a nearly linear speedup effect with the number of clients. Notably, these observations align well with our theoretical results (see Theorem 3 and Remark 5).This further validates the correctness of our theoretical analysis.

## 6    Conclusion

This paper is the first to establish the generalization error and excess error bounds for the DSGD-MGS algorithm in non-convex settings without the bounded gradients assumption. It addresses how MGS can exponentially reduce the generalization error bound and shows that even with a very large number of MGS steps, it cannot completely close the gap between decentralized and centralized training. Additionally, our theoretical results capture the impact of key factors like data heterogeneity $\delta$, communication topology spectrum $\xi$, Multiple Gossip Steps $Q$, client number $m$, and per-client data size $n$. Previous work has not unified the analysis of these critical parameters, and this paper fills that gap, offering both theoretical insights and experimental validation and significantly advancing the theoretical understanding of decentralized optimization.

**Limitation.** The theoretical findings in this paper depend on the properties of the last iteration of D-SGD in optimization theory, which is an emerging area yet to be explored. This paper derives the properties of the function value at the last iteration under the PL-condition. Future work can further explore the properties of the loss function gradient at the last iteration under non-convex conditions.

## Acknowledgment

This work is supported by STI 2030—Major Projects (No. 2021ZD0201405), NSFC Grant (No. 62576364), Shenzhen Basic Research Project (Natural Science Foundation) Basic Research Key Project (NO. JCYJ20241202124430041) and National Natural Science Foundation of China (No.62025604, No.62261160653). Miao Zhang is supported by the National Natural Science Foundation of China (NO. 62403484).

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
