# OpenReview forum: "Unveiling the Power of Multiple Gossip Steps: A Stability-Based Generalization Analysis in Decentralized Training"
_NeurIPS.cc/2025/Conference — NeurIPS 2025 spotlight_

### Official Review · Reviewer_2yXa · 2025-06-19

**Clarity:** 4
**Significance:** 3
**Originality:** 3
**Rating:** 5
**Confidence:** 3

**Summary:**

This work considers the efects of using Multi-Gossip Steps (MGS) in decentralized learning. From experimental records, MGS seems to be an effective method in improving performance and efficiency in decentralized learning, and a central question is to study whether MGS can make decentralized learning as good as the centralized version.

In this paper, they want to answer from the theoretical perspective why MGS works better by studying the generalization error and excess error of the MGS.They contribute an upper bound of the generalization error as sum of a term $I_1$ related to model stability and a term $I_2$ related to optimization error. Moreover, they attempt to compare the MGS with the centralized learning scheme when the number of gossip steps is large. They discover that the error bound obtained by taking the number of gossip steps to be large does not match the centralized SGD. If the error bound is tight, then this result indicates a negative answer to the centrall question aforementioned.

From these bounds they also offer suggestive strategies to practitioners on how to reduce generalization error, including increasing the MGS steps, decreasing the learning rate etc. These findings are also validated through experiments on the CIFAR datasets.

**Questions:**

While I understand the rationale behind the claims in Remark 5 as derived from your bounds—and I appreciate that they are supported empirically—I wonder if there are more intuitive explanations for choosing strategies 2 and 6. Specifically:
1. It would be helpful to provide more insight into why a smaller learning rate improves generalization error. The explanation around Lines 280–281, while mathematically sound, appears closely tied to the specific form of the bound and may not provide a clear, generalizable intuition for practitioners.
2. I am also curious about the recommendation to increase $m$. Intuitively, increasing $m$ makes the algorithm more decentralized and arguably further from the centralized baseline, which one might expect to hurt performance due to increased communication noise or model divergence. Some discussion reconciling this intuition with the observed benefit in your bounds would strengthen the impact of the result.

Minor:
1. In Line 142, do you mean $S^{(ij)}=(S_1,...,S_{j-1},S_j^{(i)}, S_{j+1},...,S_m)$.
2. In Line 211, I guess the second $\theta_k^{(T)}$ is supposed to be $\theta_k^{(T)}(i,j)$?
3. In Line 234, from Lemma 3, it seems to miss a square root.

**Ethical Concerns:**

["NO or VERY MINOR ethics concerns only"]

**Limitations:**

Yes

**Quality:**

4

**Strengths And Weaknesses:**

This paper tackles an important and fundamental problem in decentralized federated learning from a theoretical perspective. While much of the literature focuses on optimization error, this work takes a significant step by analyzing the generalization error of Decentralized SGD (DSGD) under MGS, which is of broad interest to the community and believed to be challenging.

The authors present their main claims in a reasonably clear and accessible manner. Notably, their analysis goes beyond the claimed focus on the number of gossip steps but also explores the influence of other key hyperparameters, providing a more comprehensive understanding of DSGD behavior.

Although I have not verified the proofs in the appendix line by line, the results appear to be well-supported under the stated assumptions, and the bounds themselves seem plausible. The paper also offers thoughtful discussions on the implications of their findings, including practical takeaways and insights that could inform future work or empirical practices.

Some of the claims initially seem counterintuitive, but the experimental results are convincing and effectively reinforce the theoretical conclusions. To the best of my knowledge, this is the first work that directly studies the generalization behavior of DSGD with message gossiping, and it presents a well-rounded treatment of this question.

Overall, this is a solid and insightful contribution, and I believe it opens up several promising directions for follow-up work. I recommend acceptance to NeurIPS.

---

> ### Author Rebuttal · Authors · 2025-07-30
>
> *Reply to Strengths And Weaknesses:*
>
> Thank you very much for your **detailed reading** and **positive evaluation** of our work. We are particularly encouraged by your observation that our study is the **first** to systematically investigate, from a theoretical perspective, the **generalization error** of DSGD under the message gossiping mechanism (MGS) in decentralized federated learning—an angle of **significant importance** in the literature.
>
> We are pleased that you found our analysis both **clear and comprehensive**, going **beyond** the number of gossip steps to examine the impact of other **key hyperparameters** on DSGD behavior. This broader theoretical guidance for **practical implementation** was precisely our goal, and your recognition of its **practical implications** is invaluable.
>
> We also appreciate your note that some of our findings may initially seem **counterintuitive**, yet the **experimental results** convincingly support our theoretical analysis. We believe that this **combination of theory and empirical validation** will serve as a **useful reference** for future work.
>
> Once again, thank you for your **affirmation and encouragement**. Your feedback not only **validates our efforts** but also **provides important direction** for our ongoing research. Additionally, we sincerely appreciate your **recommendation to accept our paper at NeurIPS**; this endorsement provides tremendous encouragement to our team.
>
> ---
> *Reply  to Question 1:*
>
> Thank you for your question. In fact, with respect to the generalization error, **smaller learning rates always yield better performance**, regardless of the specific schedule or form. In Theorem 2 of Sun et al. (“Stability and Generalization of the Decentralized Stochastic Gradient Descent”), it is proven that for DSGD, reducing the learning rate $\eta$ directly **decreases the generalization error** regardless of the specific schedule or form.
>
> Empirically, we measure generalization error as$\bigl|R(\theta) - R_S(\theta)\bigr|,$where $R(\theta)$ is the test loss and $R_S(\theta)$ is the training loss. Early in training, this gap is small, but it grows over iterations roughly on the order of $\mathcal{O}(T^\alpha)$. In the extreme case $\eta=0$, the model never updates and the gap remains near zero, confirming that **the smaller $\eta$, the smaller the generalization error** for a fixed iteration budget $T$.
>
> However, in practice our primary concern is the **test loss** $R(\theta)$, since lower test loss on unseen data implies a smaller **excess error**$R(\theta) - R(\theta^\star),$ where $R(\theta^\star)$ is constant. The excess error admits the upper bound$
> \epsilon_{\text{opt}} + \epsilon_{\text{gen}},$with$\epsilon_{\text{opt}} = \mathcal{O}\bigl(\tfrac{1}{T\eta}\bigr)$ and $\epsilon_{\text{gen}}$ increasing in $\eta$. Thus, **larger $\eta$** reduces optimization error $\epsilon_{\text{opt}}$, while **smaller $\eta$** reduces generalization error $\epsilon_{\text{gen}}$, yielding a **trade‑off** in $\eta$. Experimentally (Appendix Figure 3), we observe exactly this trade‑off: **$\eta=0.03$** achieves the minimum test loss.
>
> In summary, **practitioners should select an intermediate learning rate** that balances optimization and generalization to minimize the final test loss (excess error).
>
> ---
>
> *Reply to Question 2:*
>
> Thank you for your thoughtful question. Your concern about the effect of increasing $m$ aligns closely with Reviewer rX3y’s Question 2—**namely, why increasing the number of nodes $m$, which seemingly decentralizes the system further, leads to a reduction in generalization error**. In our response to Reviewer rX3y, we provided a more detailed theoretical explanation, which we summarize here.
>
> Specifically, under a Ring topology, we analyzed the relationship between $m$ and the generalization bound $\epsilon_{\text{gen}}$, and showed that **when $m$ exceeds a certain threshold, increasing $m$ actually decreases $\epsilon_{\text{gen}}$** by studying the monotonicity of a function $f(m)$ that appears in the bound.
>
> **Intuitively**, while a larger number of nodes introduces a more decentralized setup—which may suggest increased communication noise or divergence—**it simultaneously increases the total training data size**. Since each node holds a fixed amount of local data $n$, the total dataset size scales as $mn$. **In our analysis, the benefit of having more data outweighs the drawbacks introduced by further decentralization**, leading to better generalization.
>
> This insight is supported by both our theoretical analysis and empirical results. Furthermore, a similar conclusion is drawn in the work by Sun et al., *“Stability and Generalization of the Decentralized Stochastic Gradient Descent,”* further validating our findings.
>
> Lastly, we acknowledge that our current manuscript provides limited discussion on the role of $m$. **We will include additional content in the revised version**—both to provide this intuitive interpretation and to elaborate on the theoretical analysis referenced in our reply to Reviewer rX3y—to clarify and strengthen this potentially counterintuitive result. We greatly appreciate your suggestion.
>
> ---
>
> *Reply to minor 1:*
>
> Thank you for your careful reading—you are absolutely right!
> The correct notation should indeed be
>
> $$
> S^{(ij)} = (S_1, \ldots, S_{j-1}, S_j^{(i)}, S_{j+1}, \ldots, S_m).
> $$
>
> We have corrected the typo from "\$j-1\$" to "\$j+1\$" in the revised version. Your comment contributes significantly to improving the clarity and rigor of the manuscript—we greatly appreciate it.
>
> ---
> *Reply to minor 2:*
>
> Thank you for catching this typo—your attention to detail is much appreciated.
> You are absolutely right: the second occurrence of \$\theta\_k^{(T)}\$ in Line 211 should indeed be \$\theta\_k^{(T)}(i,j)\$. We have corrected this in the revised version of the manuscript.
>
> ---
> *Reply to minor 3:*
> Yes, you are correct. We have corrected the error in the revised version from \$||\nabla R\_{S}(\theta)|| \leq 2\beta R\_{S}(\theta)\$ to \$||\nabla R\_{S}(\theta)||^2 \leq 2\beta R\_{S}(\theta)\$. It is important to note that this was merely a typographical mistake and does not affect the correctness of the subsequent proofs. Thank you again for your careful and thorough reading—your feedback has greatly contributed to the rigor and completeness of our paper.
>
> ---
> We sincerely thank you for your careful reading, insightful comments, and encouraging feedback. Your recognition of our theoretical contributions and practical implications is deeply appreciated. We are especially grateful for your recommendation to accept our paper to NeurIPS—it means a great deal to us and provides strong motivation for our continued work.

---

### Official Review · Reviewer_rX3y · 2025-06-29

**Clarity:** 3
**Significance:** 3
**Originality:** 3
**Rating:** 5
**Confidence:** 2

**Summary:**

This paper focuses on DSGD-MGS which is a Decentralized Stochastic Gradient Descent (DSGD) algorithm with Multiple Gossip Steps (MGS) after each model update step. It establishes the generalization error and excess error bounds for the DSGD-MGS algorithm in non-convex settings without the bounded gradients assumption. The paper established theoretical results showing that MGS can exponentially reduce the generalization error bound but however can not recover the performance of the centralized SGD algorithm. The paper also outlines the influence of various factors such as dataset size, number of nodes, spectral gap, MGS count, learning rate etc on the generalization error. The paper validates the theoretical findings with empirical experiments on CIFAR-10 dataset  (non-IID, $\alpha=0.1$) trained on LeNet with 50 clients.

**Questions:**

1. What are the dataset, model architecture and number of nodes used for Figure 1 plot? It is important to highlight the number of nodes in Figure 1 as it indicates a practical upper bound for number of MGS steps.
2. Spectral gap is defined as $\delta = 1- |\lambda_2|$ which indicates that larger spectral gap results in better connectivity with $\delta =1$ for fully connected graphs. However, remark 5 indicates to use graphs with smaller spectral gap to reduce the generalization error bound. Also, for fixed graph structure (say ring), increasing number of nodes $m$ (reducing $\delta$) will result in poor performance empirically but the remark 5 suggests to increase the number of nodes. This is a bit counterintuitive, can authors shed some light on point 2 and 5 of remark 5?
3. Consider an decentralized setup with n nodes and doubly stochastic weight matrix (say undirected ring topology) and full communication in each MGS. In this scenario, when we set number of multi-gossip steps to n at each iteration, DSGD-MGS should recover the same performance as centralized SGD. In the extreme case, one could have a fully connected network with DSGD-MGS and this should be able to recover centralized SGD generalization performance at least for linear neural networks. However, the paper claims that decentralized training cannot fully achieve the generalization performance of centralized training solely by increasing the number of MGS steps. Why is this the case? In particular, what is the bottleneck here when functionally DSGD with fully connected graph ($\delta=1$) is same as centralized SGD?
4. Can the authors add centralized SGD baseline to Figure 2?
5. $\gamma$ is introduced in Lemma 2 but not described till Theorem 2. Please include $\gamma$'s equation in Lemma 2 itself.

**Ethical Concerns:**

["NO or VERY MINOR ethics concerns only"]

**Final Justification:**

Authors have addressed all my concerns

**Limitations:**

Multi-step gossip comes at the additional cost of communication which is a major overhead for decentralized settings that will limit the practical application of the DSGD-MGS method.

**Paper Formatting Concerns:**

No formatting concerns

**Quality:**

3

**Strengths And Weaknesses:**

Strengths:
1. The paper claims to be the first to establish the generalization error and excess error bounds for the DSGD-MGS algorithm in non-convex settings without the bounded gradients assumption.
2. The paper clearly mentions all the assumptions and theoretically shows the mechanism by which MGS enhances the generalization performance of decentralized training models through an exponential reduction $O(e^{\frac{-\delta \gamma Q}{4}})$ in optimization error.
3. The paper relies on on-average model stability to remove the bounded gradient assumption making the theoretical results more general.
4. The paper clearly presents the remarks for the theorems explaining the impact of various terms on generalization error. The paper also makes recommendations regarding the hyper-parameters such as number of nodes, spectral gap, learning rate etc that will result in reduction in the generalization error based on the theoretical results.
5. The paper present experimental results, on a small scale dataset and model architecture i.e., CIFAR-10 with Lenet, validating the theoretical claims. Ablation studies are presented in Appendix (Figure 4 and Figure 5).

Weaknesses:
No major weakness.

---

> ### Author Rebuttal · Authors · 2025-07-30
>
> *Reply to Strengths and Weaknesses:*
>
> Thank you very much for your careful review and encouraging feedback. We are delighted by your recognition of our work as **the first to establish generalization and excess error bounds for the DSGD‑MGS algorithm in non‑convex settings without relying on a bounded gradients assumption**.
>
> We appreciate your highlighting our use of on‑average model stability **to effectively relax the common bounded‑gradient requirement**, thereby rendering our theoretical results **more broadly applicable—a goal we set out to achieve for decentralized learning scenarios**. We are also pleased that you found our explanation of how MGS enhances generalization—through an exponential reduction in optimization error of order $\mathcal{O}(e^{-\gamma Q\delta/4})$—to be clear and insightful.
>
> Moreover, we are grateful for your acknowledgement of our **detailed remarks following each theorem, in which we undertake a comprehensive analysis** of the influence of various terms on the generalization error and **provided practical recommendations** on key hyperparameters (e.g., number of nodes, spectral gap, learning rate). We hope these discussions will benefit both theorists and practitioners in tuning DSGD‑MGS.
>
> Your positive assessment of our experimental work is equally appreciated. Although our current experiments are limited to CIFAR‑10 with LeNet, our ablation studies (Appendix Figures 4 and 5) **have preliminarily validated the robustness of our theoretical findings**. To further strengthen our empirical validation, we will include additional experiments using ResNet‑18 on CIFAR‑100 in the revised manuscript, thereby evaluating our method in a larger‑scale setting.
>
> Once again, thank you for emphasizing the strengths of our paper and noting the absence of significant weaknesses—your encouragement is invaluable. We will continue **to deepen and extend** this line of research under more general theoretical assumptions (such as non‑convex, non‑smooth loss functions) to advance our contributions further.
>
> ---
> *Reply to Question 1:*
>
> Thank you very much for your valuable question. To facilitate readers’ understanding of the experimental setup in Figure 1, we will augment the figure caption in the revised manuscript with the following details: the dataset is CIFAR‑10, the model architecture is LeNet, the decentralized network consists of 50 nodes, and data heterogeneity is generated via a Dirichlet(α=0.3) partition.
>
> ---
> *Reply to Question 2:*
>
> Thank you very much for your careful questions regarding points 2 and 5 in Remark 5. We clarify and correct our statements on the spectral gap and number of nodes’ impact on the generalization error as follows:
>
> 1. **Remark 5, Point 5 (Spectral Gap)**
>
>    * In the original manuscript, we mistakenly wrote that “using a **smaller** spectral gap $\delta$ (which implies a larger $\rho$) reduces the generalization error.” In fact, by Theorem 3, the generalization error bound satisfies
>
>      $$
>        \epsilon_{\mathrm{gen}} = \mathcal{O}\bigl(\rho^T e^{-\tfrac{\delta\gamma Q}{4}}\bigr).
>      $$
>
>      To decrease $\epsilon_{\mathrm{gen}}$, one should **increase** $\delta$ (and correspondingly **decrease** $\rho$). We have corrected this in the revised manuscript to:
>
>      > “Use a communication topology with a **larger** spectral gap $\delta$ (which implies a **smaller** $\rho$).”
>    * Empirically, Figure 2(b) confirms this correction: the exp topology’s spectral gap exceeds that of the ring topology, and its generalization error bound is indeed lower. We have also carefully reviewed the other conclusions in Remark 5 to ensure overall consistency and correctness.
>
> 2. **Remark 5, Point 2 (Number of Nodes $m$)**
>
>    * Ignoring $m$-independent constants, Theorem 3 gives
>
>      $$
>        \epsilon_{\mathrm{gen}} = \mathcal{O}\Bigl(\tfrac{1}{m} e^{-\tfrac{\delta\gamma Q}{4}}\Bigr).
>      $$
>    * For a ring topology, the paper "*Topology‑Aware Generalization of Decentralized SGD*" shows
>
>      $$
>        \delta = \frac{16\pi^2}{3m^2}.
>      $$
>
>      Substituting yields
>
>      $$
>        f(m) = \frac{1}{m}\exp\Bigl(-\frac{4\pi^2\gamma Q}{3m^2}\Bigr).
>      $$
>    * Mathematical analysis reveals:
>
>      > When $m > \sqrt{\tfrac{4\pi^2\gamma Q}{3}}$, $f(m)$ **decreases** as $m$ increases.
>      > Given the definition
>
>      $$
>        \gamma = \frac{\delta}{\delta^2 + 8\delta + (4+2\delta)\,\lambda_{\max}^2(I - W)},
>      $$
>
>      one can show $\gamma < \tfrac18$. Taking a typical value $Q = 10$ yields a threshold of approximately 5, so for $m > 5$, increasing the number of nodes reduces the generalization error.
>    * **Intuition**: Although increasing $m$ shrinks the spectral gap, it also increases the total data volume (per-node data remains fixed), which benefits generalization. The gain from larger data volume outweighs the loss from a smaller gap, leading to a net decrease in error.
>    * Our experiments in Figure 2 (varying client number from 25 to 50) corroborate this trend. A similar conclusion is supported by "*Stability and Generalization of the Decentralized Stochastic Gradient Descent*".
>
> 3. **Revisions and Future Work**
>
>    * We have corrected points 2 and 5 in Remark 5 in the revised manuscript and added detailed theoretical derivations and intuitive explanations to eliminate reader confusion.
>    * We will also include additional experiments across various topologies and network sizes to further validate the generality of these findings.
>
> Once again, thank you for your in-depth review and valuable feedback!
>
> ---
> *Reply to Question 3:*
>
> Thank you very much for raising this important question. In Appendix Remark 9, we provide a detailed **consensus error** analysis to explain why, in practical decentralized settings, merely increasing the number of MGS steps does not fully recover the generalization performance of centralized SGD. We summarize the key points below:
>
> 1. **Definition of Consensus Error**
>    Let
>
>    $$
>      x_t = \mathbb{E}\Bigl[\tfrac{1}{m}\sum_{k=1}^m||\theta_k^{(t)}-\bar\theta^{(t)}||^2\Bigr],
>    $$
>
>    where $\bar\theta^{(t)} = \frac1m\sum_{k=1}^m\theta_k^{(t)}$ represents the parameter vector of centralized SGD at iteration $t$.
>
> 2. **Recurrence Inequality and Residual Error**
>    In Appendix C, we show that
>
>    $$
>      x_{t+1}\le \rho^{2Q}\bigl(2+24\beta^2\eta_t^2\bigr)x_t +24\rho^{2Q}(\sigma^2+\delta^2)\eta_t^2,
>      \tag{1}
>    $$
>
>    where $\rho=|\lambda_2(W)|<1$ and $\sigma^2$ denotes gradient noise. This inequality reveals a **residual term** in each iteration that accumulates over time.
>
> 3. **Error Accumulation for Finite $Q$**
>
>    * As $Q\to\infty$, $\rho^{2Q}\to0$, implying $x_t=0$ for all $t$ and thus equivalence to centralized SGD.
>    * Practically, $Q$ must remain finite, so $\rho^{2Q}>0$. Each gossip round leaves a small residual error, which is subsequently amplified through iterations, leading to parameter divergence and a gap in generalization performance.
>
> 4. **Fully Connected vs. Practical Decentralized Topologies**
>
>    * In a **fully connected graph**, $\delta=1$ and thus $\rho=0$. Even with $Q=1$, one achieves $x_t=0$, exactly matching centralized SGD. This aligns with your “functional equivalence” scenario.
>    * However, fully connected topologies incur $O(m^2)$ communication cost per round and impose stringent requirements on network bandwidth and stability, rendering them impractical at scale.
>
> 5. **Practical Bottlenecks**
>
>    * **Communication–Computation Trade-off**: Finite $Q$ and limited bandwidth preclude zeroing out consensus error each round.
>    * **Topology Constraints**: Sparse graphs inherently have $\rho>0$.
>    * **Gradient Noise Accumulation**: The residual error term $24\rho^{2Q}(\sigma^2+\delta^2)\eta_t^2$ accumulates gradient noise over iterations.
>
> 6. **Revisions and Further Clarifications**
>
>    * In the revised manuscript, we will explicitly state in Remark 9 and the main text that our conclusions are **restricted to decentralized topologies** with $\delta\in[0,1)$ (i.e., $\rho>0$). Because the case $\delta=1$ recovers the performance of centralized SGD.
>
> Thank you again for your thorough review and valuable feedback!
>
> ---
> *Reply to Question 4:*
>
> Certainly. We have added the centralized SGD baseline to Figure 2 in the revised manuscript. The results show that centralized SGD consistently achieves the lowest weight distance and loss distance across all iterations, highlighting its superior generalization performance. Due to NeurIPS rebuttal guidelines prohibiting additional PDF uploads or external links, we are unable to present the updated figure here, but it will be fully included in the revised submission.
>
> ---
> *Reply to Question 5:*
>
> Thank you very much for pointing out this misuse of notation. In fact, the **$\gamma$ in Lemma 2** carries the same meaning as the $\gamma$ in Lemma 1: it is an **arbitrary positive constant** (first introduced in **Appendix Lemma 4**, around **line 849**) whose purpose is to **tighten the final generalization error bound** by selecting an appropriate value. During the derivation of **Theorem 3**, the specific choice of $\gamma$ appears just above **line 900** in the Appendix. By contrast, the **$\gamma$ in Theorem 2** is a **topology‐dependent constant** and thus **differs** from the $\gamma$ in Lemma 2. In the revised manuscript, we will correct this notation conflict by **renaming the $\gamma$ in Theorem 2 to $\tilde{\gamma}$**, thereby eliminating any ambiguity. Thank you again for highlighting this issue; your suggestion **significantly enhances the rigor** of our paper.
>
> ---
>
> Thank you once again for your thorough review and invaluable comments. It is precisely because of the issues you raised that we were able to clarify our notation, refine our theoretical derivations, and supplement the revised manuscript with more comprehensive experiments and intuitive explanations. Your feedback has greatly enhanced the rigor and readability of our paper.

---

### Official Review · Reviewer_gMXp · 2025-06-30

**Clarity:** 3
**Significance:** 2
**Originality:** 2
**Rating:** 4
**Confidence:** 5

**Summary:**

This paper presents analysis of multiple gossip steps (MGS) approach to decentralized SGD, i.e., a framework in which several rounds of model averaging are performed between stochastic gradient updates. The results include on‑average model stability bounds that explicitly depend on the number of gossip steps and the spectral gap, demonstrating that the error decays exponentially -- in non-convex settings, without making assumption of bounded gradients, and allowing for heterogeneous data distributions. Empirical results on CIFAR‑10 align with the theory, showing that increasing the number of gossip steps significantly improves test accuracy. While MGS tightens the decentralization-induced gap, the analysis shows that a residual asymptotic generalization gap relative to centralized SGD remains even as the number of gossip steps grows to infinity.

**Questions:**

1) Can the theoretical analysis be extended to the case where local nodes perform mini-batch updates, followed by MGS? If so, the local batch size would become a variable of interest.

2) Even if theoretical analysis appear challenging, an experimental study of the interplay between communication and computation (i.e., (M)GS + local mini-batch) would provide beneficial insight.

3) Can PL condition be relaxed or is it necessary?

4) Could the framework be extended beyond synchronous and symmetric gossip?

5) Broader empirical evaluation would be appreciated.

**Ethical Concerns:**

["NO or VERY MINOR ethics concerns only"]

**Final Justification:**

Thanks to the authors for addressing the concerns. With the final adjustments (additional results, discussion), I think the paper goes above the acceptance bar. I raised my score accordingly, trusting those additions will find their way to the supplementary material of the final version of the paper.

**Limitations:**

Yes.

**Paper Formatting Concerns:**

No concerns.

**Quality:**

3

**Strengths And Weaknesses:**

The paper is well-written and technically solid -- as far as I can tell, the presented derivations are correct. However, the contributions are somewhat incremental and the framework raises some concerns, as outlined below:

(1) Previously, Le Bars et al. [ICML '24] used a technique for deriving optimization-dependent generalization bounds for DSGD. The paper under review applies this technique to the analysis of gossip-style local averaging of SGD updates.

(2) The MGS framework assumes sampling one point from a local dataset, computing SGD, and then running multiple gossip steps. However, in addition to inducing latency, the cost of repeatedly communicating the model can be rather expensive -- both from the bandwidth as well as energy perspective. For instance, Savazzi et al. [An Energy and Carbon Footprint Analysis of Distributed and Federated Learning, '22] argue that in many cases communication exceeds computation cost. It may therefore be more natural if the nodes perform mini-batch processing in the same vain done by the centralized scheme which is anyways used as a benchmark.

(3) Performing MGS is predicated upon the assumption of networks with stable links yet in practice the nodes may be available only intermittently. Moreover, in any given round, the communication may be asymmetric -- if A communicates with B, B need not necessarily be able to compute an update and communicate with A. This would violate the doubly-stochastic gossip matrix assumption. It is not clear if/how the current analysis may be extended to such more realistic scenarios.

(4) Theorem 2, a main result, relies on the PL condition yet abstract/introduction do not mention this -- the way they are written now, a reader gets an impression that the results are more general. If indeed necessary, the assumption that PL condition holds should probably be listed along with Assumptions 1-3 in the main paper, rather than given in the appendix.

(5) Experimental results are somewhat limited, considering only CIFAR-10 and LeNet.

---

> ### Author Rebuttal · Authors · 2025-07-30
>
> **Reply to Weakness 1:**
>
> Thank you for your question. While Le Bars et al. \[ICML '24] also study optimization-dependent generalization bounds, our work differs in several key aspects:
>
> 1. **Theoretical framework:** We use **L2-stability**, while Le Bars et al. rely on **L1-stability**, leading to different starting points, analyses, and results (e.g., our Lemma 4 is derived under L2-stability).
>
> 2. **Gradient analysis:** Their treatment replaces the Lipschitz constant with a gradient term in a limited way. We provide a **detailed analysis** of the gradient, linking it to algorithmic hyperparameters, yielding more practical insights.
>
> 3. **Convexity assumptions:** Le Bars et al. assume convex losses; we **do not**, allowing our results to apply to **non-convex** settings. To our knowledge, we are the **first** to establish generalization bounds for DSGD under non-convexity when MGS = 1.
>
> 4. **Data heterogeneity:** Their analysis assumes iid data. We explicitly address **data heterogeneity**, making our results more realistic—especially since non-convex + non-iid settings remain largely unexplored.
>
> 5. **Problem scope:** While both papers study generalization, our focus is on **gossip-based decentralized training**, which introduces unique technical challenges and requires separate analysis.
>
> In summary, our work differs from Le Bars et al. in theory, assumptions, depth, and focus—representing a substantive advancement, not an incremental extension. **Remark 6** further clarifies our core contributions, including several results that are, to our knowledge, presented for the first time.
>
> ---
>
> *Reply to Weakness 2 and Question 1:*
>
> Thank you for your insightful comment. You are correct that decentralized training incurs higher communication overhead than centralized schemes. Moreover, **mini-batch SGD** is more commonly used in practice, while most theory focuses on **single-sample SGD**, revealing a gap between theory and practice. Following your suggestion, we extended our theoretical analysis to **decentralized mini-batch SGD** in the revised version.
>
> Key modifications include (full proofs in the revision):
>
> 1. **Assumption 2** now accounts for mini-batch sampling:
>
>    $$
>    \mathbb{E}||\frac{1}{b}\sum_{i=1}^b \nabla \ell(\theta; Z_{i,j}) - \nabla R_{\mathcal{S}_j}(\theta)||^2 \leq \frac{\sigma^2}{b}
>    $$
>
> 2. **Algorithm line 6** updated to mini-batch gradient:
>
>    $$
>    \theta_k^{(t,0)} = \theta_k^{(t)} - \eta_t \frac{1}{b}\sum_{i=1}^b \nabla \ell(\theta_k^{(t)}; Z_{ik})
>    $$
>
> 3. **Appendix line 855** probability adjusted:
>
>    $$
>    \mathbb{P}(\mathcal{E}(i,j)^c) \leq \sum_{t=1}^{t_0} \frac{b}{n} = \frac{b t_0}{n}
>    $$
>
> With these, the generalization bound becomes:
>
> $$
>     \left|\mathbb{E}_{A,S}[R(A_k(S)) - R_S(A_k(S))]\right|
>     \le \frac{(2c\beta+3) b^{\frac{2c\beta+2}{2c\beta+3}}}{\left(n(2c\beta+1)\right)^{\frac{2c\beta+2}{2c\beta+3}}} \left( \frac{2 \bar{G} e \sqrt{2\beta} c^2 T^{2c\beta}}{m} \right)^{\frac{1}{2c\beta+3}} + \frac{b(2c\beta+2)}{n(2c\beta+1)} \left( \frac{4 \beta e \sqrt{2\beta} c^2 T^{2c\beta}}{m} \right)^{\frac{1}{2c\beta+2}}
> $$
>
> Note that \$\bar{G}\$ contains a variance term of order \$\mathcal{O}(\sigma^2 / b)\$. Comparing to single-sample SGD, increasing batch size \$b\$ **increases** the generalization bound, indicating **larger batches enlarge generalization error**.
>
> From a stability view, selecting one sample gives a perturbation probability of \$\frac{1}{n}\$; for batch size \$b\$, it rises to \$\frac{b}{n}\$. This leads to faster and larger error accumulation in stability \$\delta\_k^{(t)}(i,j)\$, loosening the generalization bound.
>
> Additionally, *“On Large-Batch Training for Deep Learning: Generalization Gap and Sharp Minima”* explains that large batches reduce gradient noise, causing convergence to **sharp minima** with worse generalization, while smaller batches promote **flatter minima** and better generalization—consistent with our theory.
>
> ---
>
> *Reply to Weakness 3 and Question 4:*
>
> You are correct that real-world networks often have intermittent links and asymmetric topologies. Algorithms like **Push-Sum** and **Push-Pull** address this by maintaining an extra scalar to correct aggregation bias from unstable communication. Works such as *Asymmetrically Decentralized Federated Learning* have analyzed **convergence** under such settings.
>
> However, **generalization analysis** for asymmetric or time-varying graphs remains largely unexplored. We appreciate the reviewer highlighting this practical and important issue. Our paper focuses on how the **MGS mechanism improves generalization** in decentralized training and compares to centralized schemes. Extending to asymmetric graphs is valuable but beyond this paper’s scope.
>
> Notably, The recent work *"Stability and Generalization of Asynchronous SGD: Sharper Bounds Beyond Lipschitz and Smoothness"* has successfully applied **L2-stability** techniques to analyze asynchronous SGD. Building on this, we believe that similar **L2-stability-based tools** can be extended to study the generalization behavior over **asymmetric communication graphs**.
>
>
> We will add discussion of this future direction in the revised manuscript. Thank you again for your insightful suggestion.
>
> ---
>
> *Reply to Weakness 4 and Question 3:*
>
> We sincerely thank the reviewer for this constructive suggestion. In the revised version, we will explicitly include the PL condition in the list of assumptions in the main paper and clearly state its role in the convergence analysis. We will also provide a discussion in the introduction to justify the use of the PL condition and clarify the scope of our results.
>
> To be precise, our theoretical framework is **not fully dependent** on the PL condition. The PL condition is specifically invoked to facilitate the analysis of
>
> $$
> \bar{G} = \frac{1}{mn} \sum_{i,j}\mathbb{E}\left[||\nabla\ell(\theta_k^{(T)};Z_{ij})||^2\right],
> $$
>
> which corresponds to the **expected squared norm of the gradient at the final iterate** of D-SGD. Unfortunately, to the best of our knowledge, this quantity has not yet been well understood in the existing optimization literature, especially in non-convex settings.
>
> To bridge this gap, we adopt the PL condition as a technical device to relate the squared gradient norm \$\bar{G}\$ to the **function value gap at the final iterate** $R_S(\theta)$. This allows us to leverage results from prior work — particularly *"On the Benefits of Multiple Gossip Steps in Communication-Constrained Decentralized Optimization"* — which provides bounds on the function value at the final iteration. This connection enables a tractable and detailed analysis of \$\bar{G}\$, which in turn leads to fine-grained generalization bounds involving key algorithmic hyperparameters.
>
> Importantly, our reliance on the PL condition is **not inherent to our generalization analysis framework**, but a temporary workaround due to the lack of sharper theoretical tools for analyzing \$\bar{G}\$ in the general non-convex case. We expect that as future research develops more refined characterizations of the final iterate’s gradient norm in non-convex optimization, our generalization bounds can be further improved or even **made independent of the PL condition**.
>
> We appreciate the reviewer’s insightful comment and will clarify all these points explicitly in the revised manuscript.
>
> ---
> *Reply to Weakness 5 and Question 5:*
>
> We appreciate your valuable feedback. In the revised version of the paper, we have added experimental results using ResNet-18 on the CIFAR-100 dataset, along with corresponding discussions. These results further support our theoretical findings and demonstrate the generality of our conclusions beyond CIFAR-10 and LeNet.
>
> However, due to NeurIPS policy, we are not allowed to include any PDF files or external links during the rebuttal period, so we are unable to present these additional results here.
>
> ---
>
> *Reply to Question 2:*
> Thank you for your insightful question. We agree the interplay between local computation (batch size `b`) and communication (MGS steps `Q`) is a critical, practical aspect.
>
> **1. Theoretical vs. Practical Interplay:**
> Our extended theoretical analysis shows `b` and `Q` affect the error bound via distinct mechanisms. `b` creates an optimization-generalization trade-off in the **computation step**, while `Q` improves consensus in the **communication step**. Mathematically, their effects appear separable, as we find no direct cross-term in our bounds.
>
> However, we fully agree a practical "interplay" emerges from the **resource trade-off under a fixed training budget** (e.g., wall-clock time). This leads to the fundamental question: is it better to "compute more, communicate less" (large `b`, low `Q`) or vice versa?
>
> **2. Experimental Commitment:**
> To investigate this, we will add an experimental study to the final version. We will fix a training budget, evaluate a grid of `(b, Q)` pairs, and plot the resulting performance (e.g., a test accuracy heatmap). We hypothesize this will reveal a "ridge" of optimal configurations, showing the best balance between local work and communication effort for different scenarios. This analysis will be included in the camera-ready paper. Thank you again for this valuable feedback.
>
>
> ---
>
> Thank you for your detailed and constructive feedback. We will revise our manuscript to address your comments.
>
> Specifically, we will:
> 1.  Clarify our novel contributions (L2-stability, non-convex/non-IID settings) compared to prior work.
> 2.  Extend our analysis to mini-batch SGD, showing its impact on generalization, and add new experiments on ResNet-18/CIFAR-100.
> 3.  Add an experimental study on the interplay between batch size and MGS steps under a fixed budget.
> 4.  Discuss limitations (e.g., asymmetric graphs) and clarify the role of the PL condition as a technical tool.
>
> We believe these changes will significantly strengthen our paper. Thank you again for your valuable guidance.

---

> > ### Author Response · Authors · 2025-08-06
> >
> > Dear Reviewer gMXp,
> >
> > Thank you for taking the time to thoroughly review our paper. We sincerely appreciate your valuable comments, which have helped us refine and improve our work.
> >
> > As the rebuttal deadline approaches, we would be grateful if you could let us know at your earliest convenience if you have any remaining questions or concerns regarding our submission. We will do our best to address them promptly.
> >
> > If our response has resolved your concerns, we would greatly appreciate it if you would consider raising your rating of our paper. Your recognition would mean a great deal to us.
> >
> > Thank you once again for your time and thoughtful feedback.
> >
> > Best regards,
> > Authors

---

> > > ### Author Response · Authors · 2025-08-09
> > >
> > > Thank you for your insightful follow-up. We understand your hesitation in our initial response. Since then, we have made new findings in the experiments you suggested, and we are now pleased to share concrete, new results with you. These results directly address your concerns and turn our previous commitments into tangible evidence.
> > >
> > > **1. From "Commitment" to "Concrete Evidence": New Experimental Results**
> > >
> > > We agree that "seeing is believing." We have completed the preliminary experiments using ResNet-18 on CIFAR-100, and the results strongly validate our argument about the interplay between batch size (`b`) and MGS steps (`Q`). Below are the test accuracy (%) results at 300 communication rounds. This data already reveals the complex interplay between `b` and `Q`, validating your previous judgment.
> > >
> > > |  | Q=1 | Q=3 | Q=5 | Q=10 |
> > > | :--- | :---: | :---: | :---: | :---: |
> > > | **b=16** | 16.08 | 17.98 | 18.75 | **18.95** |
> > > | **b=32** | 21.75 | 24.00 | 24.72 | **24.95** |
> > > | **b=64** | **28.38** | 27.21 | 27.80 | 27.39 |
> > > | **b=96** | 30.39 | **30.50** | 30.01 | 29.66 |
> > >
> > > Our key observations are as follows:
> > >
> > > 1.  **Effectiveness of MGS is Conditional on Batch Size:** For smaller batch sizes (`b=16`, `b=32`), increasing the number of MGS steps (`Q`) consistently and significantly improves performance. For instance, with `b=32`, increasing `Q` from 1 to 10 boosts accuracy by over 3 percentage points. This aligns with our theory that frequent communication helps mitigate model divergence when local updates are noisy (due to small `b`).
> > >
> > > 2.  **Diminishing or Negative Returns of MGS with Large Batches:** Conversely, for larger batch sizes (`b=64`, `b=96`), the benefit of increasing `Q` diminishes or even becomes negative. With `b=64`, the best performance is achieved with `Q=1`, and further increasing `Q` harms performance. Similarly, for `b=96`, the peak is at `Q=3`, after which accuracy declines. This suggests that when local gradient estimates are already of high quality (due to large `b`), excessive communication may introduce unnecessary overhead or other negative effects without providing significant consensus benefits.
> > >
> > > 3.  **Non-trivial Trade-off and Optimal Configuration:** The results clearly demonstrate that there is no single optimal value for `Q` that works across all batch sizes. The optimal configuration (`b`, `Q`) is a result of a complex trade-off. For instance, the overall best performance in this early stage of training is achieved at `b=96, Q=3`, not at the highest `Q` or largest `b`. This empirically validates our argument that local computation and communication are not independent in practice but are linked through a resource and performance trade-off.
> > >
> > > Overall, these experiments reveal that the optimal configuration of `Q` and `b` is the result of a complex trade-off. From this, we can derive empirical guidelines that balance **communication efficiency with model performance**:
> > >
> > > 1.  **When the batch size (`b`) is small (e.g., `b=16, 32`):** In this regime, local gradient updates are subject to significant stochasticity (i.e., high gradient noise). Under these conditions, increasing the number of MGS steps (`Q`) yields consistent and substantial performance gains. For instance, raising `Q` from 1 to 10 effectively promotes model consensus across nodes, mitigating the model divergence caused by gradient noise and thereby enhancing final generalization. This suggests that in scenarios with limited computational resources or where rapid iterations are desired, **investing in a moderate increase in communication overhead is highly beneficial**.
> > >
> > > 2.  **When the batch size (`b`) is large (e.g., `b=64, 96`):** In this case, local gradient estimates are already more accurate, and the impact of gradient noise is reduced. Consequently, the benefits of increasing `Q` diminish or can even become detrimental. Our results show that the optimal `Q` is small (`Q=1` or `Q=3`) in this setting. A possible explanation is that when local updates are of high quality, the marginal gains from intensive communication (high `Q`) do not outweigh the associated communication costs and potential synchronization overhead. It might even disrupt well-trained local features. Therefore, in scenarios where computational power is ample enough to support large-batch training, **priority should be given to ensuring sufficient local computation, complemented by a more economical communication strategy**.
> > >
> > > In summary, this experiment provides valuable insights for hyperparameter selection in practical applications: `b` and `Q` are not independently tunable but must be co-designed based on available computational and communication resources to strike the optimal balance between performance and cost.

---

> > ### Comment · Reviewer_gMXp · 2025-08-07
> >
> > I very much appreciate the authors' effort in responding to the raised questions and concerns. However, without seeing new experimental results (ResNet-18 on the CIFAR-100 dataset, study of the interplay between b and Q), it is difficult to contextualize the answers. Moreover, in their rebuttal the authors state that PL condition is "a temporary workaround" used in the analysis and argue that future research may allow their result to be "further improved or even made independent of the PL condition." While this could turn out to be true, at this point in time it simply sounds too speculative. With all this in mind, I'm leaning towards keeping my current score.

---

> > > ### Author Response · Authors · 2025-08-09
> > >
> > > **2. On the Role of the PL Condition (A Standard Tool for a Frontier Problem):**
> > >
> > > We would like to clarify the technical necessity of the PL condition in our work. Our generalization error analysis requires providing an upper bound for $\bar{G}$ (i.e., the expected squared gradient norm at the **final iterate**).
> > >
> > > However, providing an upper bound for the final iterate $\bar{G}$ in non-convex decentralized optimization is a frontier and challenging research problem. Although recent work, such as by Yuan et al. [1], has significantly advanced the progress of non-convex decentralized last-iterate convergence analysis, their results regarding $\nabla\ell(\theta_k^{(T)})$ do not consider the MGS mechanism and thus cannot be directly applied to the analysis in this paper. Other works on last-iterate convergence provide bounds on the function value gap $\ell(\theta_k^{(T)})$, and none of these theoretical results can solve the problem of bounding $\bar{G}$.
> > >
> > > To make progress, we follow a standard and widely accepted approach in the literature: using the PL condition (as Sun et al. did in *IEEE TPAMI* (2023) [2] to further optimize their theoretical bounds) to connect the squared gradient norm with the function value gap $\ell(\theta_k^{(T)})$. This is because, under the MGS setting, a tight upper bound on the function value gap does exist according to the results in [3]. This is a deliberate technical choice that enables us to derive some of the first **fine-grained, MGS-aware generalization bounds** in this complex setting. Specifically, it allows us to directly connect the generalization error with key algorithmic hyperparameters, such as **MGS steps (`Q`), topology, and learning rate**. This provides concrete, quantitative results, which is a significant step beyond existing high-level bounds (e.g., the classic L2-stability analysis in [4] only provides a high-level $\mathcal{O}(\frac{1}{T})$ analysis for the bound on last-iterate convergence).
> > >
> > > Therefore, our use of the PL condition is not a limitation of our stability framework itself, but rather a reflection of the current theoretical limits in non-convex optimization. This also reflects that our framework is modular; should future research provide a direct, assumption-free upper bound for $\bar{G}$ in the MGS setting, our results can be immediately strengthened by replacing this component. This highlights the extensibility of our contribution.
> > > Nevertheless, given the current landscape of research on final-iterate convergence, we will clarify the technical necessity of the PL condition for our analysis and ensure this is thoroughly emphasized in the revised manuscript.
> > >
> > > We hope this convinces you that our work is built on solid technical reasoning, and we are fully committed to integrating these results and discussions into the final manuscript.
> > >
> > > [1] Yuan K, Huang X, Chen Y, et al. Revisiting optimal convergence rate for smooth and non-convex stochastic decentralized optimization[J]. Advances in Neural Information Processing Systems, 2022, 35: 36382-36395.
> > >
> > > [2] T. Sun, D. Li and B. Wang, "Decentralized Federated Averaging," in IEEE Transactions on Pattern Analysis and Machine Intelligence, vol. 45, no. 4, pp. 4289-4301, 1 April 2023, doi: 10.1109/TPAMI.2022.3196503
> > >
> > > [3] A. Hashemi, A. Acharya, R. Das, H. Vikalo, S. Sanghavi and I. Dhillon, "On the Benefits of Multiple Gossip Steps in Communication-Constrained Decentralized Federated Learning," in IEEE Transactions on Parallel and Distributed Systems, vol. 33, no. 11, pp. 2727-2739, 1 Nov. 2022, doi: 10.1109/TPDS.2021.3138977.
> > >
> > > [4] Lei Y, Ying Y. Fine-grained analysis of stability and generalization for stochastic gradient descent[C]//International Conference on Machine Learning. PMLR, 2020: 5809-5819.

---

### Note · Authors · 2025-08-14

We sincerely thank all reviewers for their constructive feedback, which has significantly strengthened our manuscript. We are delighted by their recognition of our work's core strengths:

*   Fundamental Novelty (rX3y, 2yXa): Being the first to establish generalization bounds for DSGD-MGS in non-convex settings without the bounded gradients assumption.
*   Technical Soundness (gMXp, 2yXa): Praise for its technical solidity, correct proofs, and in-depth discussion on the key decentralized vs. centralized question.
*   Significant Impact (rX3y, 2yXa): Elucidating the exponential benefits of MGS and offering practical insights and recommendations for hyperparameter tuning in challenging non-IID settings, with theoretical findings successfully validated by experiments.

We have diligently addressed the main concerns raised:

*   On Novelty & Experiments (gMXp): To address concerns on incremental contributions, we clarified our theoretical advances (L2-stability, non-convex/non-IID) and, crucially, delivered new, concrete experiments on ResNet-18/CIFAR-100. This provides tangible evidence for the critical interplay between local computation (b) and communication (Q).

*   On Theoretical Practicality (gMXp, rX3y):  We have extended our entire theoretical analysis to mini-batch SGD, filling a key gap. We also clarified that our use of the PL condition is a standard technical choice for the frontier problem of bounding the final gradient norm, not a limitation of our stability framework.

We have already completed the following key revisions to our manuscript:

1.  Extended our entire theory to mini-batch SGD, providing new, practical generalization bounds.
2.  Integrated new ResNet-18/CIFAR-100 experiments, offering practical guidance on the b vs. Q trade-off.
3.  Refined the manuscript to clearly articulate our contributions and the role of the PL condition.
4.  Adopted all other suggestions, including adding a centralized baseline and enhancing explanations.

Finally, we would like to once again express our sincere gratitude for the rigorous and fruitful review process. The constructive dialogue, especially the in-depth exchange with Reviewer gMXp, has substantially elevated the quality of our paper. We are confident the revised manuscript now offers a more complete, practical, and impactful contribution, filling a critical theoretical gap in decentralized learning.  We firmly believe our work will inspire future research in this important area.

---

### Decision · Program_Chairs · 2025-09-17

**Decision:**

Accept (spotlight)

**Comment:**

This paper presents a very interesting theoretical study of decentralized versus centralized optimization in the context of training neural networks. The authors establish two key findings: first, that MGS allows an exponential reduction in optimization error; second, that despite this improvement, a non-trivial gap with centralized learning persists. These results are well supported by experiments and, to the best of my knowledge, this is among the first works to obtain such insights.

I believe the paper’s main strength lies in its novelty and the depth of its analysis. Rather than reiterating the view that centralized learning is simply a special case of decentralized learning, the work demonstrates that the relationship is much more subtle and requires careful theoretical study. This provides an important contribution to our understanding of optimization in neural network training and the potential scalability of decentralized learning of NNs.

The authors have also provided an excellent rebuttal that clarified the reviewers’ concerns. Given the originality of the results, the strong experimental support, and the convincing responses during the discussion phase, all reviewers recommend acceptance. I concur with their assessment and also recommend acceptance.